# Autophagy: The Potential Link between SARS-CoV-2 and Cancer

**DOI:** 10.3390/cancers13225721

**Published:** 2021-11-16

**Authors:** Parham Habibzadeh, Hassan Dastsooz, Mehdi Eshraghi, Marek J. Łos, Daniel J. Klionsky, Saeid Ghavami

**Affiliations:** 1Research Center for Health Sciences, Institute of Health, Shiraz University of Medical Sciences, Shiraz 71348-14336, Iran; Parham.Habibzadeh@yahoo.com; 2Department of Life Sciences and Systems Biology, University of Turin, Via Accademia, Albertina, 13, 10123 Torino, Italy; Hassan.Dastsooz@unito.it; 3IIGM-Italian Institute for Genomic Medicine, c/o IRCCS, Candiolo, 10126 Torino, Italy; 4Candiolo Cancer Institute, FPO-IRCCS, 10060 Torino, Italy; 5Department of Human Anatomy and Cell Science, Rady Faculty of Health Sciences, Max Rady College of Medicine, University of Manitoba, Winnipeg, MB R3E 0J9, Canada; eshraghi.mehdi@gmail.com; 6Biotechnology Center, Silesian University of Technology, 44-100 Gliwice, Poland; 7Life Sciences Institute, University of Michigan, Ann Arbor, MI 48109, USA; klionsky@umich.edu; 8Research Institute of Oncology and Hematology, Cancer Care Manitoba, University of Manitoba, Winnipeg, MB R3E 0V9, Canada; 9Faculty of Medicine, Katowice School of Technology, ul. Rolna 43, 40-555 Katowice, Poland

**Keywords:** colorectal neoplasms, COVID-19, gastrointestinal neoplasms, immune checkpoint inhibitors, neoplasms, oncogenic viruses, oncolytic virotherapy, post-acute COVID-19 syndrome, reactive oxygen species, tumor escape

## Abstract

**Simple Summary:**

Coronavirus disease 2019 (COVID-19) has led to a global crisis. With the increasing number of individuals infected worldwide, the long-term consequences of this disease have become an active area of research. The constellation of symptoms COVID-19 survivors suffer from is commonly referred to as post-acute COVID-19 syndrome in the scientific literature. In this paper, we discuss the potential long-term complications of this infection resulting from the persistence of the viral particles in body tissues interacting with host cells’ autophagy machinery in the context of the development of cancer, cancer progression and metastasis, as well as response to treatment. We also propose a structured framework for future studies to investigate the potential impact of COVID-19 infection on cancer.

**Abstract:**

COVID-19 infection survivors suffer from a constellation of symptoms referred to as post-acute COVID-19 syndrome. However, in the wake of recent evidence highlighting the long-term persistence of SARS-CoV-2 antigens in tissues and emerging information regarding the interaction between SARS-CoV-2 proteins and various components of the host cell macroautophagy/autophagy machinery, the unforeseen long-term consequences of this infection, such as increased risk of malignancies, should be explored. Although SARS-CoV-2 is not considered an oncogenic virus, the possibility of increased risk of cancer among COVID-19 survivors cannot be ruled out. Herein, we provide an overview of the possible mechanisms leading to cancer development, particularly obesity-related cancers (e.g., colorectal cancer), resulting from defects in autophagy and the blockade of the autophagic flux, and also immune escape in COVID-19 survivors. We also highlight the potential long-term implications of COVID-19 infection in the prognosis of patients with cancer and their response to different cancer treatments. Finally, we consider future directions for further investigations on this matter.

## 1. Introduction

Coronavirus disease 2019 (COVID-19), caused by severe acute respiratory syndrome coronavirus 2 (SARS-CoV-2), has led to unprecedented mortality and morbidity at a global scale. SARS-CoV-2 is a positive-stranded RNA virus belonging to the *Coronaviridae* family, members of which interact with different components of the cellular autophagy machinery [1,2]. A constellation of symptoms, such as fatigue, exhaustion, and shortness of breath, persisting long after the resolution of the acute phase of infection, has been reported among some survivors of this infection [3]. The persistence of symptoms beyond 12 weeks after the initial onset is called post-COVID-19 syndrome [3,4]. However, long-term complications of this infection could extend well beyond these symptoms.

## 2. Autophagy and Cancer

Autophagy plays a prominent role in maintaining cellular homeostasis through the removal of damaged organelles, abnormal proteins, and invading organisms. Defects in autophagy are associated with various pathological conditions, including cancer [5]. It can lead to accumulation of damaged mitochondria and alter cellular metabolism, leading to a high oxidative state [6]. Furthermore, impairments in autophagy flux can lead to ER stress and subsequent accumulation of chaperone proteins and an eventual rise in the unfolded protein burden [7,8]. This chronic injury to various cellular organelles and proteins and the accumulation of the genetic damage in cells with defective autophagy can lead to the development of pathophysiologies, as evidenced by the detection of loss-of-function mutations in different autophagy genes in various malignancies such as colorectal cancer [9]. However, despite this significant role of autophagy in the initiation of tumors, it is a double-edged sword in cancer, and its role is highly context dependent [10]. Cancer cells are dependent on the cellular autophagy machinery due to their rapid growth and biosynthetic demands [6,8]. Autophagy also plays a significant role in promoting metastasis in certain tumors, particularly in RAS-driven cancers [11]. Conversely, autophagy can also be leveraged to enhance the response to various cancer treatments [12] (Figure 1).

## 3. Potential Clues and Mechanisms Supporting the Role of SARS-CoV-2 in Oncogenesis

Several oncogenic viruses exert their carcinogenesis through altering autophagy [13]. Although the oncogenic potential of SARS-CoV-2 has not yet been investigated, two other positive-sense single-strand RNA viruses, namely, hepatitis C virus/HCV and human T-cell lymphotropic virus type 1/HTLV-1, exploit the cellular autophagy machinery in order to cause liver cancer and adult T-cell leukemia/lymphoma/ATLL, respectively [13].

Both ACE2 (the major SARS-CoV-2 receptor) and TMPRSS2 (a transmembrane serine protease necessary for viral cell entry) display a very high level of expression in the human gastrointestinal tract [14]. This virus can infect and actively replicate in human enterocytes [14]. A recent study on the gastrointestinal biopsies of 14 individuals performed four months after their COVID-19 diagnosis reported persistence of the viral nucleic acids and antigens in 50% of the cases [15]. Persistent shedding of SARS-CoV-2 viral particles for months after the resolution of the symptoms has also been reported in the nasopharyngeal specimens obtained from individuals with previous infection [16,17]. This continued presence of the viral antigens in different tissues could have serious long-term consequences for survivors of COVID-19 infection.

Interestingly, various SARS-CoV-2 proteins interact with different components of the cellular autophagy pathway (Figure 2). For example, viral ORF3a protein interacts with VPS39, a part of the homotypic fusion and protein sorting/HOPS complex, leading to the inhibition of the fusion of autophagosomes with lysosomes [18]. In addition, SARS-CoV-2 Nsp15 blocks the induction of autophagy, whereas the viral ORF7a protein decreases the acidity of lysosomes, which can interfere with autophagosome–lysosome fusion as well as cargo degradation [19]. Moreover, accumulation of SQSTM1 and an increase in the processed LC3B (LC3B-II) levels have been observed upon SARS-CoV-2 envelope (E) protein overexpression [19,20]. Altogether, these findings indicate that various SARS-CoV-2 antigens block autophagic flux in the infected cells. Conversely, a recent study showed that the protein encoded by viral ORF8 leads to major histocompatibility complex I (MHC-I) degradation in the affected cells by targeting these molecules for lysosomal degradation through the BECN1-dependent autophagy pathway [21].

The persistence of the SARS-CoV-2 antigens in the enterocytes could lead to long-term defects in the cellular autophagy machinery, as evidenced by recent studies. The blockage in the autophagic flux in these cells would lead to the accumulation of SQSTM1/p62 (an autophagic receptor protein), reactive oxygen species, organelle damage, and genetic alterations in response to different stresses, eventually leading to tumorigenesis [22,23]. An approximate 1.5-fold increase in SQSTM1 levels in the presence of SARS-CoV-2 ORF3a, ORF7a, or E proteins has been observed [19]. SQSTM1 plays a significant role in tumor transformation due to its important function as a signaling molecule interacting with many oncogenic pathways, including those involving NFE2L2/NRF2 and NFKB/NF-κB [24]. Several studies also reported evidence of mitochondrial dysfunction, excessive production of reactive oxygen species, ER stress, and unfolded protein responses in the cells infected with SARS-CoV-2 [25,26,27]. As mentioned, these pathological events could be, in part, a result of disturbed autophagy flux in the cells. Furthermore, disrupted cell cycle regulation due to autophagy defects would lead to the uncontrolled proliferation of cells carrying defective genetic materials, paving the way for the development of cancer [28]. In addition, activation of compensatory mechanisms in colorectal cancer cells in specific contexts secondary to the inhibition of autophagy lead to tumor growth [29]. This could in turn lead to a higher risk of cancer development and more rapid proliferation of tumor cells among COVID-19 survivors.

Downregulation of MHC-I is one of the major mechanisms of the immune evasion by tumor cells through escaping detection by CD8^+^ T-cells and their cytotoxicity [30]. This adaptive immune escape mechanism is observed in many malignancies, including colorectal cancer, and is associated with a poor prognosis [31]. MHC-I degradation through the BECN1-dependent autophagy pathway induced by viral ORF8 could, therefore, provide a fertile soil for carcinogenesis by blunting the ability of the immune system to detect the cancer cells because their neo-antigens are no longer presented on the MHC-I molecules.

It should be noted that SARS-CoV-2 antigens have also been detected in various other human organs, such as the lungs, heart, kidneys, hepatobiliary system, and the lymphatic system [32]. Although most of these findings have been reported in post-mortem studies and the persistence of the viral antigens in these organs among COVID-19 survivors has not so far been assessed due to technical and ethical considerations, it is likely that viral particles could linger in these tissues, continuing to interact with the host cell autophagy machinery and therefore inducing carcinogenesis in various organ systems. Chronic SARS-CoV-2 infection is reported in immunocompromised patients (e.g., due to anti-cancer therapy) and animal models [33,34,35]. These immunocompromised patients with chronic infection might have a higher risk of cancer, as evidenced in other malignancies with suspected infectious etiology [36].

Interestingly, a recent study has reported that SARS-CoV-2 RNA can integrate into the genome of cultured human cells through reverse transcription. The authors also claimed that they were able to detect viral-host chimeric transcripts in the patient-derived tissues, suggesting that these transcripts are a result of integration of DNA copies of viral sequences in the human genome [37]. However, a later study by another group did not find any evidence for such an event in human cells [38]. Clearly, further large-scale studies are required to investigate if the SARS-CoV-2 genome is able to integrate into the human genome.

A nation-wide population-based study conducted in Denmark investigating mortality rates among patients admitted to the hospitals for non-COVID-19 diseases during the pandemic from March 2019 to January 2021, and found a consistently higher mortality rate among patients with cancer compared with baseline pre-pandemic mortality rates among these patients [39]. Although this observation is likely due to a multitude of factors, it is possible that defects in autophagy and increased tumor immune evasion among patients with certain malignancies who had previously contracted this infection could have led to more a rapid progression of their cancer due to the processes described above. In addition, a 6.9-fold increase in tumor burden has been reported in patients who had been diagnosed with metastatic colorectal cancer after the first lockdown compared with those diagnosed prior to the lockdown [40]. Despite the significant role of the delays in the screening and diagnosis as a result of the COVID-19-related lockdown, the above-mentioned processes could have also contributed to this observation. Notably, patients with cancer remain a vulnerable population for COVID-19 [41].

## 4. Autophagy and Metabolism

Autophagy plays a fundamental role as a catabolic process recycling intracellular components into breakdown products that could be used in various cellular metabolic pathways [42]. This, in turn, enables cells to survive under metabolic stress conditions (e.g., nutrient deprivation, hypoxia, etc.) [43,44].

Alteration in cellular metabolism is one of the hallmarks of cancer [45]. Unlike normal cells, which mainly rely on mitochondrial oxidative phosphorylation for energy production, most cancer cells depend on aerobic glycolysis (the Warburg effect) [45,46]. Despite being an inefficient pathway for adenosine 5′-triphosphate (ATP) generation, it is thought that it confers an advantage to tumor cells in incorporating nutrients in tumor biomass and cellular proliferation [46]. Considering the diverse role of autophagy in feeding different metabolic pathways through degradation of various cellular substrates, it is not surprising that elevated basal autophagy is seen in many tumor types [8,47].

Furthermore, mitochondrial metabolism plays an important role in various tumor types by redox balance, ATP generation, and synthesis of intermediates required for macromolecule biosynthesis (e.g., nucleotides) [48]. The selective autophagic elimination of the mitochondria (i.e., mitophagy) is a very important cellular process regulating their number and also participating in mitochondria quality control [49]. In addition, mitophagy plays a vital role in the metabolic rewiring of cancer cells aimed at meeting their bioenergetic needs [50].

Considering the paramount role of autophagy in maintaining tissue homeostasis, dysregulation in this process has been linked to cancer development and progression in a context-specific manner [51]. The potential inhibition of the autophagy flux by SARS-CoV-2 antigens can deprive tumor cells of the building blocks essential for unconstrained tumor proliferation and could, therefore, limit tumor growth. However, the accumulation of dysfunctional organelles, particularly dysfunctional mitochondria, as a result of impaired autophagy can pave the way for the development of cancer [52] (Figure 3).

Obesity is correlated with elevated systemic oxidative stress [53]. The excessive nutrients supply, overwhelming the cellular Krebs cycle and mitochondrial respiratory chain, leads to mitochondrial dysfunction and increases the formation of reactive oxygen species (ROS) [54]. Elevation of intracellular ROS is known to lead to the upregulation of the cellular autophagic response, which subsequently removes defective mitochondria and therefore limits the generation of ROS [55,56,57]. The potential inhibition of autophagic flux as a result of the persistence of SARS-CoV-2 antigens in different tissues could blunt this protective mechanism and could particularly be more important in the pathogenesis of obesity-related cancers. ROS levels are higher in colorectal cancer cells compared with normal non-cancerous tissues [58]. Furthermore, ROS play a critical role in mediating tumorigenesis and colorectal cancer initiation driven by RAC1 [59]. The effects of ROS extend beyond the initiation of obesity-related cancers; they are anti-apoptotic factors promoting the survival of pancreatic cancer cells [60].

## 5. Implications for Cancer Treatment

The long-term presence of SARS-CoV-2 antigens in cancer tissues could also have wide-ranging implications for cancer therapy. MHC-I loss or downregulation is a common mechanism for the development of resistance to PDCD1/PD-1 inhibitors among patients with melanoma [61,62]. The same phenomenon could potentially render cancer cells arising in the presence of SARS-CoV-2 antigens resistant to anti-PDCD1 monotherapy as a result of MHC-I degradation through BECN1-dependent autophagy. This outcome could necessitate the use of PDCD1 and CTLA4 blockade combination therapy in such cases, as this treatment strategy does not require MHC-I expression for exerting its therapeutic effects [63].

Lysosomal sequestration of weak base hydrophobic chemotherapeutic agents decreasing their accessibility to their target sites and the resultant decrease in their cytotoxic effects is a significant challenge that culminates in treatment failure and subsequent increase in mortality due to cancer [64]. Lysosomal exocytosis facilitating the clearance of chemotherapeutics accumulated in this organelle is another important component of lysosome-mediated chemotherapy resistance [65]. A recent study has shown that SARS-CoV-2 ORF3a not only inhibits the autophagy flux but also promotes lysosomal exocytosis [66]. This could confer drug resistance to the cancer cells arising in the setting of persistence SARS-CoV-2 ORF3a presence.

Despite the preliminary clues pointing towards the role of the virus in promoting tumor progression described here, rare instances of tumor burden reduction in three patients with colorectal cancer and disease remission in a patient with Hodgkin lymphoma following COVID-19 infection have been reported in the literature [67,68]. Although the over-exuberant immune response instigated by SARS-CoV-2 attacking the tumor cells could have led to this phenomenon [69], the complex role of autophagy in cancer (acting as a double-edged sword) should not be underestimated in interpreting these exceptional cases. These serendipitous observations may partly be due to the blockage of autophagy flux, leading to the deprivation of cancer cells of the essential biosynthetic materials generated by the cellular autophagy machinery required for tumor growth [6,8]. This in turn highlights the therapeutic potential of SARS-CoV-2 in oncolytic virotherapy via blockage of autophagic flux in specific tumor types at certain stages of their natural course. Theoretically, this potential therapeutic strategy could particularly be effective against tumor cells with high autophagy activity.

## 6. Future Directions

Notwithstanding significant efforts to unravel the role of autophagy in COVID-19 infection, many questions, particularly those surrounding the role of autophagy in the long-term complications of COVID-19, remain unanswered [70,71]. A multidisciplinary approach, involving both clinical and basic science researchers, aimed towards studying the unforeseen long-term consequences of COVID-19 infection in cancer development, tumor progression, metastasis, and response to various cancer therapeutics could eventually piece together different parts of this puzzle, providing a better understanding of the complex interaction between SARS-CoV-2 and cancer.

The potential long-term oncogenic effects of SARS-CoV-2 antigens that inhibit the autophagic flux (e.g., NSP15, ORF3a, etc.) could be investigated through in vitro studies assessing changes in tumor formation following the long-term presence of these antigens in various human cell lines, such as human *KRAS* knockout cells. In addition, studies on patient-derived cancer cell lines could provide valuable insight into the effect of viral antigens and the subsequent blockade of the autophagic flux and MHC-I downregulation on tumor cell response to various cancer treatments and its effect on metastatic potential. Furthermore, although due to the short lifespan of most animal models they may not be suitable for investigating the long-term effects of viral antigens on cancer development, patient-derived mouse xenograft models of different human malignancies that have arisen in the presence of SARS-CoV-2 antigens can provide an ideal framework for exploring tumor growth, metastasis, and the response to different cancer treatments. Comprehensive molecular phenotyping of various human tumors arising in the setting of long-term presence of SARS-CoV-2 antigens and comparing them with malignancies developed in other settings could shed light on the potential biological processes specifically affected (Figure 4).

Clinical and epidemiological investigations can also play a principal role in expanding our understanding of this matter. Cohort studies with sufficient follow-up can assess the long-term incidence of various malignancies in different populations and would determine whether prior COVID-19 infection is a risk factor for the development of different cancers adjusting for well-known cancer risk factors. In addition, studying specific subpopulations, such as those with severe infection (i.e., ICU admission), and also patients with primary and secondary immunodeficiency could offer profound insight into the most susceptible populations. Considering the potential far-reaching implications of the infection for patients with active cancer, specific studies focused on this vulnerable population assessing the long-term impact of the infection on the clinical course of their malignancy should be carried out. Prospective studies investigating if prior COVID-19 infection affects the rate of cancer progression, mortality rate, and response to treatment among patients with different malignancies, adjusting for the traditional contributing factors, could provide insight into the possible interaction between SARS-CoV-2 antigens and neoplastic cells (Figure 5).

## 7. Conclusions

Despite the short amount of time since SARS-CoV-2 was first reported in 2019, scientists around the globe have managed to unravel various aspects of this infectious disease. However, we are still far from a solid understanding of this emerging infection and many important questions, particularly those regarding the long-term complications of this disease, remain unanswered.

In this work, building on the previous investigations demonstrating long-term persistence of the SARS-CoV-2 nucleic acids and antigens in human tissues and also other research studies showing the interaction of the viral particles with the host autophagy machinery, we hypothesize that SARS-CoV-2 could potentially be an oncogenic virus by blocking the autophagic flux, and also leading to immune escape by downregulation of MHC-I. We also propose that the resultant dysregulation in cellular autophagy could affect the response to treatment in cancer cells. Further laboratory-based, clinical, and population-based studies are required to explore this matter.

## Figures and Tables

**Figure 1 cancers-13-05721-f001:**
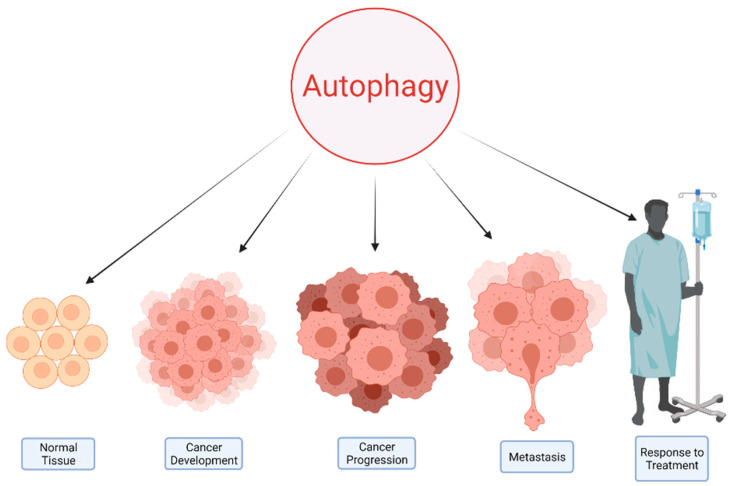
Autophagy has a principle role in both the health and disease states, such as malignancies. This process plays a complex role in various aspects of cancer.

**Figure 2 cancers-13-05721-f002:**
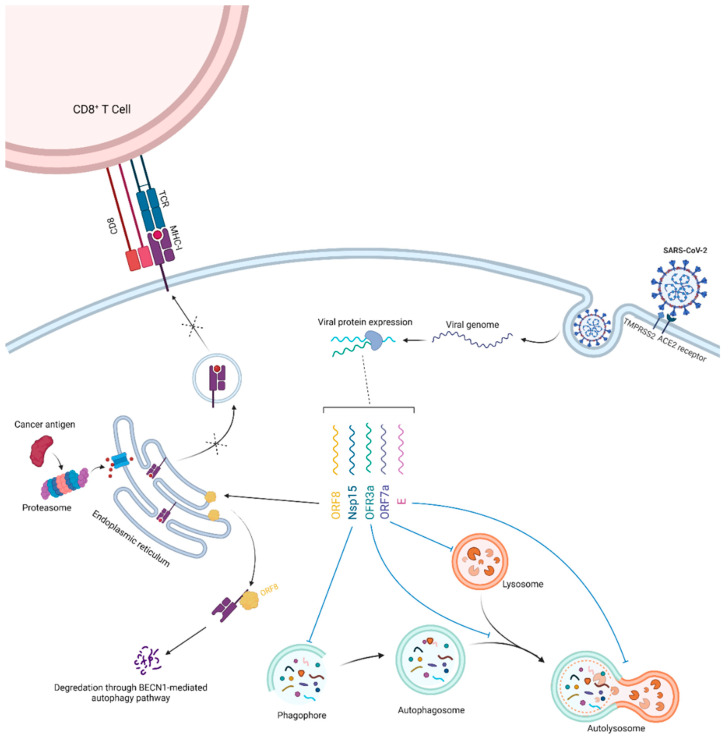
SARS-CoV-2 infects enterocytes through ACE2 receptors and TMPRSS2 expressed on their surface. The release of the viral genome and production of viral proteins (simplified in this schema) will lead to the interaction of various SARS-CoV-2 antigens with the host cell autophagy machinery. Overall, this will induce the blockage of the autophagic flux. The accumulation of reactive oxygen species, damaged cellular proteins and organelles, and acquired genetic defects due to various stressors can lead to the development of malignancies. Furthermore, infection can also interfere with the cellular MHC-I antigen-presentation pathway, blunting the ability of host cytotoxic CD8^+^ T cells to recognize potential oncogenic antigens.

**Figure 3 cancers-13-05721-f003:**
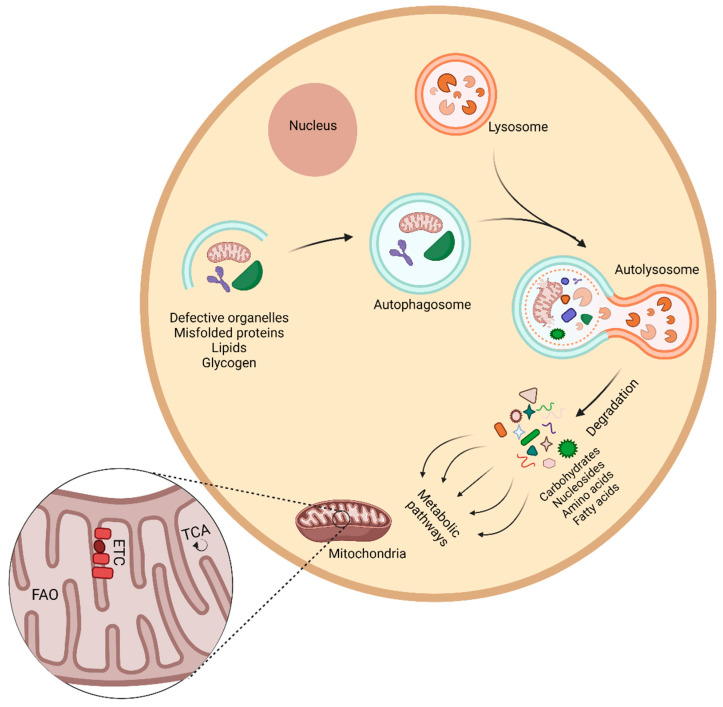
Autophagy and cell metabolism in cancer. Autophagy plays an important role in removing defective cellular organelles and macromolecules and recycling them for further use in different metabolic pathways. Mitochondria are the primary cellular organelles responsible for various cellular metabolic pathways including fatty acid oxidation (FAO), the tricarboxylic acid (TCA) cycle, and the electron transport chain (ETC). Impaired production of metabolic intermediates important for cellular proliferation can lead to a metabolic crisis and subsequently limit tumor growth. Conversely, excessive production of reactive oxygen species and the resultant oxidative damage can lead to tumorigenesis.

**Figure 4 cancers-13-05721-f004:**
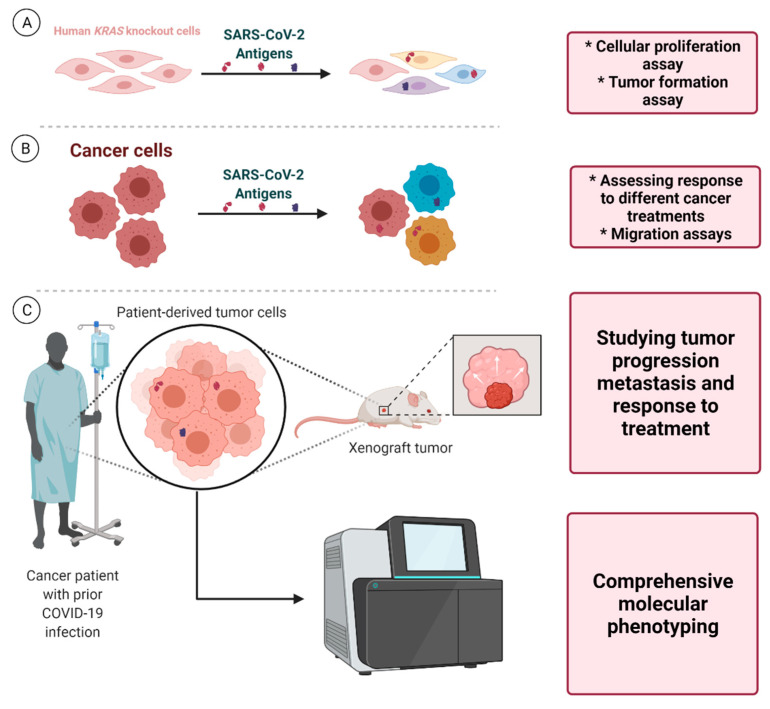
Proposed framework for laboratory-based investigations. (**A**) Investigations that could unravel the potential effects of various SARS-CoV-2 antigens on cellular proliferation and tumor formation in human KRAS knockout cells. (**B**) Using patient-derived cancer cells, the effect of these antigens and the subsequent autophagy blockade and MHC-I downregulation on response to different cancer treatments and metastasis (migration assays) could be studied. (**C**) Furthermore, studying tumor specimens arising in the presence of SARS-CoV-2 antigens using xenograft mouse models and also advanced molecular phenotyping techniques could unravel tumor biological behavior (growth rate, metastasis, etc.), its molecular characteristics, and the underlying factors affecting its response to various cancer therapeutics.

**Figure 5 cancers-13-05721-f005:**
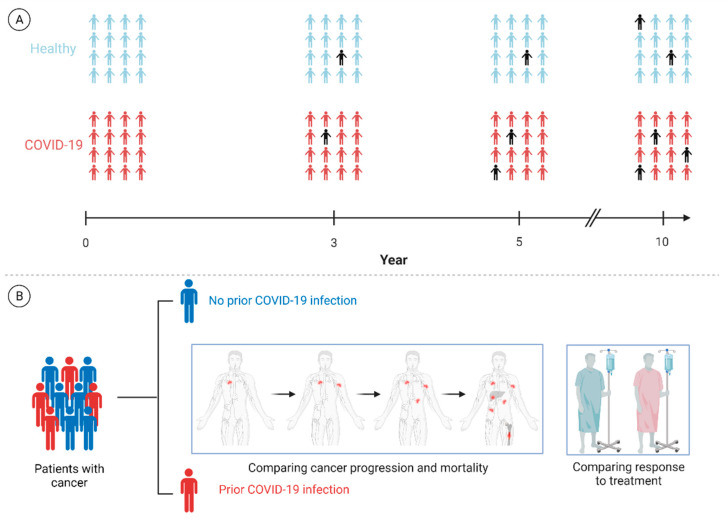
Proposed framework for clinical and population-based investigations. (**A**) Studies that can investigate whether prior COVID-19 infection is a risk factor for development of malignancies in subsequent years. (**B**) Other investigations could also shed light on the potential long-term effects of SARS-CoV-2 antigens present in cancer tissues on cancer progression and mortality rate and the response to different cancer therapeutics. The back icon in (**A**) represents a patient with a new diagnosis of cancer. Please note that the numbers of individuals with cancer in both groups have been arbitrarily chosen and are not based on real-life data. Future investigations will hopefully examine the hypothesis described here.

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
