# Peer review of "Autophagy: The Potential Link between SARS-CoV-2 and Cancer"

_cancers, 2021, doi:10.3390/cancers13225721_

Round 1
Reviewer 1 Report
The current paper provide a very valuable hypothesis about the impact of Autophagy in Obesity related cancers. It is a timely paper which will be very helpful for the field There are few comment to consider: 1- I suggest the authors provide ide/hypothesis (in a paragraph) if autophagy could be important in risk of COVID-19 induced other cancers too. We all know autophagy is directly related to cell metabolism and cancer could be considered a metabolic disease. So it is important other cancers are also discussed in a short section/paragraph 2- I recommend the authors provide a scheme about the impact of autophagy in metabolism and its impact on regulation of it so the readers understand it better.Author Response
- “I suggest the authors provide idea/hypothesis (in a paragraph) if autophagy could be important in risk of COVID-19 induced other cancers too. We all know autophagy is directly related to cell metabolism and cancer could be considered a metabolic disease. So it is important other cancers are also discussed in a short section/paragraph.”
We have now added a section entitled “Conclusion” (page 10, lines 338-353) and have now provided a summary of our hypothesis in a paragraph which highlights its broader scope beyond obesity-related cancers. Considering this valuable comment, we have also updated the manuscript title in order to encompass all malignancies.
- “I recommend the authors provide a scheme about the impact of autophagy in metabolism and its impact on regulation of it so the readers understand it better.”
We thank you very much for highlighting this point. We have now added a new scheme to the manuscript demonstrating the role of autophagy in metabolism (figure 3) and have also provided a more comprehensive overview of the interactions between autophagy, cell metabolism, and cancer in section 4 (pages 5-7, lines 187-241).
Reviewer 2 Report
In this paper, the authors presented the hypothesis of a link between COVID-19 and cancer. They reported an overview of possible mechanisms leading to cancer development, and cancer treatment failure resulting from defects in autophagy in COVID-19 survivors.
The main hypothesis reported are:
- SARS-CoV-2 antigens persist in cells and tissues of COVID-19 survivors – particularly in individuals (obese patients) / cells (enterocytes) displaying a very high expression level of ACE2 and TMPRSS2
- SARS-CoV-2 antigens have some effects on cellular proliferation and tumor formation via autophagy (particularly in RAS-driven cancers / obesity related cancers)
- SARS-CoV-2 antigens could change the response to different cancer treatments and metastasis formation via autophagy blockade and MHC-I downregulation
This paper is very interesting, and alerts the medical community on potential SARS-CoV-2 carcinogenesis through altering autophagy. Figures are beautiful and informative.
I have the following comments:
- My main concern is about the organization of the manuscript:
- The simple summary and the abstract do not report the main ideas of the paper
- The link between cancer and autophagy is well described, the hypotheses on covid and autophagy are correctly presented, however, the link with obesity is not well reported... The part “Autophagy and Obesity” must be developed or perhaps withdrawn… It will allow a better balance between parts and a better readability of hypotheses.
- Overall, a better organization of the manuscript is absolutely mandatory to increase the clarity of the manuscript.
- The term “COVID-19” have to be changed for “SARS-CoV-2” in the title
- Authors have to better explain the different presentations of COVID-19. Most people who have COVID-19 recover completely within a few weeks (ad integrum). However, some people, even those who had mild disease, continue to experience symptoms after their initial recovery. These conditions have been called post-COVID-19 syndrome or long COVID-19. In these patients several symptoms persist but SARS-CoV-2 RT-PCR are negatives in respiratory and blood samples. Conversely, in some immunocompromised patients (particularly in patients treated with anti-CD20 and cancer therapy) SARS-CoV-2 RT-PCR are positives in respiratory and blood samples for weeks or months. These patients have persistent (or chronic) COVID-19. Much is still unknown about howCOVID-19 will affect people over time. Nevertheless, the persistence of the viral antigens in organs among COVID-19 survivors is more likely in patients with chronic COVID-19, and may be long COVID-19 than in those who have recovered ad integrum… A statement should be added to moderate the scope of the hypotheses.
Author Response
- “My main concern is about the organization of the manuscript:
The simple summary and the abstract do not report the main ideas of the paper.”
We have now revised the summary and abstract to include the main ideas reported in the paper.
- “The link between cancer and autophagy is well described, the hypotheses on covid and autophagy are correctly presented, however, the link with obesity is not well reported... The part “Autophagy and Obesity” must be developed or perhaps withdrawn… It will allow a better balance between parts and a better readability of hypotheses.”
We agree with the reviewer’s point regarding this matter and have, therefore, changed this section to “Autophagy and Metabolism” and subsequently provided a comprehensive overview of the interaction of autophagy, cancer, and metabolism (pages 5-7, lines 187-241). We have, furthermore, added a schema to further demonstrate the material covered in this section (figure 3).
- “Overall, a better organization of the manuscript is absolutely mandatory to increase the clarity of the manuscript.”
We have now modified the organization of the manuscript per this valuable recommendation. Specifically, the part on “Autophagy and Obesity” is further developed and is now renamed to “Autophagy and Metabolism” as described above. In addition, a new section entitled, “Conclusion”, has also been added to the text (page 10, lines 338-353).
- “The term “COVID-19” have to be changed for “SARS-CoV-2” in the title.”
We agree with the reviewer and have now updated the title.
- “Authors have to better explain the different presentations of COVID-19. Most people who have COVID-19 recover completely within a few weeks (ad integrum). However, some people, even those who had mild disease, continue to experience symptoms after their initial recovery. These conditions have been called post-COVID-19 syndrome or long COVID-19. In these patients several symptoms persist but SARS-CoV-2 RT-PCR are negatives in respiratory and blood samples. Conversely, in some immunocompromised patients (particularly in patients treated with anti-CD20 and cancer therapy) SARS-CoV-2 RT-PCR are positives in respiratory and blood samples for weeks or months. These patients have persistent (or chronic) COVID-19. Much is still unknown about howCOVID-19 will affect people over time. Nevertheless, the persistence of the viral antigens in organs among COVID-19 survivors is more likely in patients with chronic COVID-19, and may be long COVID-19 than in those who have recovered ad integrum… A statement should be added to moderate the scope of the hypotheses.”
We think that the reviewer has raised an important point for consideration. We have now added a statement to the Introduction in order to provide a clear definition of post-COVID-19 syndrome (page 2, lines 56-57) and have also highlighted the potential risk in chronic SARS-CoV-2 infection as already evidenced in other malignancies with suspected infectious etiology based on a meta-analysis (page 5, lines 160-161). Accordingly, considering the important point the reviewer highlighted, we added a statement regarding further avenues for clinical and population-based research (page 9, lines 317-320).
We were pleased to see that Reviewer #1 commented, “The current paper provides a very valuable hypothesis about the impact of Autophagy in Obesity related cancers. It is a timely paper which will be very helpful for the field.” Similarly, we appreciate the comments of Reviewer #2, “This paper is very interesting, and alerts the medical community on potential SARS-CoV-2 carcinogenesis through altering autophagy. Figures are beautiful and informative.” The comments have allowed us to improve our manuscript, which we hope is now acceptable for publication.
Round 2
Reviewer 2 Report
I thank the authors for the corrections they have made to their manuscript. The message is more precise and clear.
One last remark: please, change the following sentence in the introduction by adding "some": "A constellation of symptoms such as fatigue, exhaustion and shortness of breath persisting long after the acute phase of the infection has resolved has been reported among some survivors of this infection".
Thank you again for the opportunity to read your interesting work.
Author Response
We have addressed the reviewer’s comments as follows (note that new text in the revised manuscript is in red font):
Reviewer 2:
I thank the authors for the corrections they have made to their manuscript. The message is more precise and clear.
One last remark: please, change the following sentence in the introduction by adding "some": "A constellation of symptoms such as fatigue, exhaustion and shortness of breath persisting long after the acute phase of the infection has resolved has been reported among some survivors of this infection".
We would like to thank you again for your constructive comments which improved the quality of the manuscript substantially. We have now addressed your concern in the mentioned statement and have revised the sentence accordingly (page 2, line 56). Thank you.